# Characteristics of Microplastics and Their Affiliated PAHs in Surface Water in Ho Chi Minh City, Vietnam

**DOI:** 10.3390/polym14122450

**Published:** 2022-06-16

**Authors:** Nguyen Thao Nguyen, Nguyen Thi Thanh Nhon, Ho Truong Nam Hai, Nguyen Doan Thien Chi, To Thi Hien

**Affiliations:** 1Faculty of Environment, University of Science, 227 Nguyen Van Cu Street, District 5, Ho Chi Minh City 700000, Vietnam; ngtnguyen@hcmus.edu.vn (N.T.N.); nttnhon@hcmus.edu.vn (N.T.T.N.); htnhai@hcmus.edu.vn (H.T.N.H.); ndtchi@hcmus.edu.vn (N.D.T.C.); 2Faculty of Environment, University of Science, Vietnam National University, Ho Chi Minh City 700000, Vietnam

**Keywords:** microplastics, PAHs, surface water, chemical composition, Ho Chi Minh City

## Abstract

Microplastic pollution has become a worldwide concern. However, studies on the distribution of microplastics (MPs) from inland water to the ocean and their affiliated polycyclic aromatic hydrocarbons (PAHs) are still limited in Vietnam. In this study, we investigated the distribution of MPs and PAHs associated with MPs in canals, Saigon River, and Can Gio Sea. MPs were found at all sites, with the highest average abundance of MPs being 104.17 ± 162.44 pieces/m^3^ in canals, followed by 2.08 ± 2.22 pieces/m^3^ in the sea, and 0.60 ± 0.38 pieces/m^3^ in the river. Fragment, fiber, and granule were three common shapes, and each shape was dominant in one sampling area. White was the most common MP color at all sites. A total of 13 polymers and co-polymers were confirmed, and polyethylene, polypropylene, and ethylene-vinyl acetate were the three dominant polymers. The total concentration of MPs-affiliated PAHs ranged from 232.71 to 6448.66, from 30.94 to 8940.99, and from 432.95 to 3267.88 ng/g in Can Gio sea, canals, and Saigon River, respectively. Petrogenic sources were suggested as a major source of PAHs associated with MPs in Can Gio Sea, whereas those found in Saigon River and canals were from both petrogenic and pyrogenic sources.

## 1. Introduction

In the 1950s, plastic was first manufactured and was considered as one of the most important inventions because its outstanding properties (durable, waterproof, easy to mold...) bring convenience to people [1]. However, plastics are non-biodegradable; they have a massive number of adverse effects on the environment and organisms. Plastic trash persisting in the environment is subjected to physical, chemical, and biological impacts, and would be fragmented into pieces smaller than 5 mm—known as microplastics (MPs) [2]. Additionally, MPs are produced directly as a primary source in cosmetics, skin care products, and resin pellets [3]. After being discharged into water bodies, MPs can cause gastrointestinal obstruction through ingestion, reduced mobility to organisms through entanglement, and enter the food chain through bioaccumulation [4,5,6]. In addition, with large surface areas and hydrophobic properties, MPs easily adsorb heavy metals and persistent organic pollutants (POPs). They enhance toxicity in the aquatic environment, sea creatures, and even human beings [7,8,9], adversely enhancing the level of ecological risk [10,11]. Polycyclic aromatic hydrocarbons (PAHs)—one among the foremost common POPs—are ubiquitous within the environment and have been well-concerned long-time ago because of their carcinogenic and mutagenic risk [12,13].

Much research has been carried out since the 2000s to understand the behavior and fate of MPs [2,14]. Initial studies on protocols for the monitoring and distribution of MPs shows that MPs were detected in many places [15,16,17], even in Antarctica [18]; therefore, MPs have become a global concern. Particularly, the water environment has gained the most attention, because this is often the source of more than 80% of plastic garbage [19]. However, the reported studies mainly specialize in the physical and chemical characteristics of MPs; the information of potential toxic pollutants adhered to MPs are limited, especially PAHs. PAHs in MPs may be from manufacture or adsorbed from the environment [20,21]. The study of Van [20] found high concentrations of PAHs on unexposed commercial polystyrene foam (240–1700 ng/g). PAHs are well-known because of their toxicity, carcinogenicity, and mutagenicity [22,23]. Moreover, MPs can adsorb and concentrate PAHs from the aquatic environments, then transfer and bioaccumulate through the food chain and finally affect human health [10,20,24]. After long-term exposure in the aquatic environment, concentrations of PAHs sorbed to MPs can be many times higher than those in the aqueous environment [25].

In recent years, the number of investigations on MPs in inland water has increased. MPs in canals, rivers were reported in various studies [26,27,28]. Inland water is the main source contributing to the MPs in the ocean; especially when pollutants adhere to MPs surface, human health and ecosystem are more seriously affected [8]. Therefore, there is a need for further studies on the distribution of MPs in the inland water environment and the ability to absorb toxic contaminants.

Vietnam is one of top countries discharging plastic wastes into the ocean, and in recent years, MPs pollution in Vietnam has been paid more attention. MPs have been detected in different media, including surface water [29,30,31], sediment [32,33], marine organisms [34], and even in the atmosphere [35,36]. A high abundance of MPs in the Saigon River in Ho Chi Minh City was concluded in Lahens’ investigation [29]. However, the published research was only conducted locally—in one sampling area such as a river or canals, there is a lack of studies on the presence of MPs in the outflow from the inland to the receiving source (marine environment). In Vietnam, previous studies have found a remarkable level of PAHs in water bodies specifically in sediment in HCMC [37,38]. The concentration of a total of 16 PAHs in sediment in Saigon River ranged from 49 ng/g to 933 ng/g dw, with the dominance of Phe, Flt, Pyr, and BbF [36]. MPs are a chemical carrier and can penetrate the food web, therefore, PAHs in MPs can bioaccumulate in the organisms [10,24,37]. There is a lack of study on the sorption of POPs in general and PAHs specifically by MPs in Vietnam. There should be more studies on the presence of MPs in surface water, from the sources to the ocean, as well as the pollutants that adhere to MPs.

Ho Chi Minh City (HCMC)—the most populous city in Vietnam—is the leading economic, cultural, and industrial center with a population of more than 10 million people [38]. Saigon River flows through HCMC with systems of canals weaving in the urban area. They receive the direct discharge of wastewater and other anthropogenic activities, then follow the current of the Saigon River to the Can Gio Sea. Therefore, in our study, we selected three main canal systems, Saigon River, and Can Gio Sea of HCMC to investigate MPs and their affiliated PAHs. The objectives of this study are: (1) to provide information on the abundance and distribution of MPs in surface water in canals, river, and sea in HCMC; (2) to characterize physical and chemical composition of MPs; (3) to investigate the distribution of 14 PAHs affiliated to MPs and their potential sources.

## 2. Materials and Methods

### 2.1. Study Area and Sample Collection

In this study, MPs in surface water were collected in canals, Saigon River, and Can Gio sea in HCMC. A total of 45 sampling sites were set along three canals, Saigon River, and Can Gio Sea in August 2020. The details on location and sampling sites are shown in Figure 1 and Appendix A.

There are three main canal systems in HCMC, these being Tan Hoa-Lo Gom (TH-LG), Nhieu Loc-Thi Nghe (NL-TN), and Tau Hu-Ben Nghe (TH-BN). These three canals belong to the network of waterways and drainage system of HCMC. TH-LG, with a length of 7.84 km, passes through 4 urban districts. NL-TN is in the city center with a length of 9.47 km, flowing through 5 districts. TH-BN is a large tributary (25.4 km in length) of Saigon River in the south of the city center, flowing through 5 urban districts. Along these canals, there are many residential areas and markets with different anthropogenic activities. Five sampling sites were chosen from the beginning to the end of each canal, evenly located along the canal, meaning a total of 15 sampling sites were chosen for three canals. At each site, 480 L of surface water (depth: 0–50 cm) was collected by using a stainless-steel bucket during the high tide of the day [39,40,41].

Saigon River (flows along the territory of HCMC, about 80 km) is the downstream of these canals; it is also the main waterway for ships entering and leaving Saigon harbor. After leaving the inner city, Saigon River merges with Dong Nai River and divides it into tributaries that flow into Can Gio Sea. Can Gio—a suburban district—is bordered by large estuaries, such as Soai Rap, Dong Tranh, Long Tau, and Nga Bay, transferring water through an area of muddy Sac forest and flowing into 2 bays of Dong Tranh and Ghenh Rai, and then the water flows into the sea. The trawling method was applied to collect microplastic samples from surface water on Saigon River and Can Gio Sea [42,43]. A total of 15 sampling sites of 5 evenly spaced sections of the Saigon River were chosen. The 15 sampling sites in the Can Gio Sea belong to four different communes of Can Gio and were about 500 m from the shore, and the sampling area included the estuarine zone of rivers from inland. A hydro-bios microplastic trawl with a mesh size of 330 µm was attached to the vessel’s side, and samples were retained in the cod end. The sampling time was from 10 to 15 min at a speed of 3 knots. A half of the trawl net’s open end was immersed under the water.

Each sample was divided into 2 parts for MPs characteristics analysis and PAHs investigation. The sample was then sieved directly through the 5 mm and 0.5 mm sieves in situ, and the upper part of the 0.5 mm sieve was rinsed and transferred directly to the brown glass bottles. They were kept at 4 °C and transported to the laboratory for further analysis. Samplers (buckets, sieves, trawl) were washed carefully by distilled water to eliminate the contaminants before the next sampling. In this study, the unit of microplastic abundance is shown as pieces per cubic meter (pieces/m^3^).

### 2.2. Sample Preparation for Physical Characteristics and Chemical Composition Analysis of Microplastics

The microplastic extraction method is referenced from previous studies with some suitable modifications [1,27,44,45]. General principles include sieving, wet oxidation, floatation, and filtration steps. After sieving in situ, at laboratory, samples were wet oxidized with H_2_O_2_ and Fe(II) solution as catalyst to remove organic compounds adhered to MPs surface. Then density separation was applied to float MPs with a mixture of NaCl and ZnCl_2_ solution (d = 1.6 g/mL). The solution was filtered through Whatman 0.45 µm filter paper to retain the supernatant. The filter paper finally was dried and observed under the microscope to determine characteristics of MPs.

In this study, polymer types of MPs were confirmed by using FTIR-ATR, an apparatus that can measure MPs from 0.5 mm in diameter. Therefore, we focus on analyzing MPs from 0.5 mm to 5 mm in diameter. The number, size, shape, and color of MPs were identified by an embedded digital microscope (MicroCapture Plus, Mustech Electronics Co., Ltd., Shenzhen, China). The number of MPs in the post-treated sample was determined by counting in three replicates. The sample was also dimensioned to know the size distribution of the MPs. MPs were also classified according to the primary colors. Shapes of MPs were classified based on their morphological features. MPs can be detected in many different shapes depending on the appearance and the formation of MPs. MPs could be fragmented due to the environmental effects (waves, UV light,...) or they were manufactured with the specific shapes to suit production purposes, and even created from synthetic fabric clothes [1]. Therefore, when observing the appearance of MPs, we classified fragments as smaller parts originating from larger pieces with irregular shapes. Fiber was defined as fibrous plastic that might come from clothes, fishing nets, etc. Another typical type that we found was granule (more three-dimensional particles); some had specific shapes (cylindrical or spherical) and white translucent color (commonly). These might have been produced as industrial resin pellets, and their appearance could change a bit depending on their state of weathering [6].

The MPs’ chemical compositions were analyzed using FTIR Spectrometer (FT/IR-6600 (Jasco—Hachioji, Tokyo, Japan) wavelength 497.544–4003.5 cm^−1^; resolution 4 cm^−1^; 32 scans/spectrum). The device was equipped with an attenuated total reflection (ATR) single-point reflector configured with a ZnSe crystal with an incident angle of 45°, it was used to transmit the reflected infrared beam to the detector. The MPs piece with dimensions larger than 0.5 mm was placed on the ATR crystal for determination, the anvil lowered to provide good contact between MPs and the optical crystal. After each sample run, we rescanned the background with 32 scans to ensure no interference. Spectra Manager™ Suite software was used to interpret the IR spectrum of the sample. The results were compared with a standard spectrum database [43].

### 2.3. Analysis of PAHs in Microplastics

In this study, the procedure for PAHs determination was followed by the previous study of Tan [24]. After being immersed in H_2_O_2_ for 24 h to remove natural organic matter, MPs were filtered through a glass microfiber filter (Whatman GF/B). Then, the filter with MPs was placed in a desiccator at least 24 h to dry and weighed before chemical analysis. After that, PAHs associated with MPs were ultrasonically extracted in 20 mL of n-hexane three times. The extracts were combined and concentrated to 1 mL using a rotary evaporator. In the next step, the concentrated solution was then transferred to a silica gel (70–230 mesh) column. For purification, the silica gel columns were added with 5 mL hexane, 30 mL hexane, respectively, and eluted with 20 mL hexane/dichloromethane (3/1, *v*/*v*). After elution, the extract that contains PAHs was evaporated until it was about 1 mL and dried by a gentle stream of nitrogen. A total of 1 mL of methanol was added to dissolve PAHs. Finally, PAHs were analyzed using HPLC with a fluorescence detector (Shimadzu, Japan). The 14 targeted PAHs in this study are: acenaphthalene (Ace), acenaphthylene (Acy), fluorene (Flu), phenanthrene (Phe), anthracene (Ant), fluoranthene (Flt), pyrene (Pyr), benz[a]anthracene (BaA), chrysene (Chr), benzo[b]fluoranthene (BbF), benzo[k]fluoranthene (BkF), benzo[a]pyrene (BaP), dibenz[a,h]anthracene (DahA), indeno[1,2,3-cd]pyrene (InP), and benzo[g,h,i]perylene (BghiP).

### 2.4. Quality Assurance and Quality Control

Throughout the whole process of sampling and analysis in the laboratory, it was that cross contamination of samples and exposure of MPs in the air was avoided. Various measures were applied, such as all investigators being equipped with cotton coats, gloves, and masks; also, only glass and metal apparatus were used during the experimental process and were washed carefully. The samples were covered by aluminum foil to avoid contamination. In addition, blank controls were also conducted using the same procedure of sample.

A mix standard of 16 PAHs (EPA 610 PAHs) was purchased from Supelco/Sigma-Aldrich (Bellefonte, PA, USA)for making standard curve and spike solutions. However, in this study, we only analyzed 14 PAHs. The blank samples were treated as the field samples to determine blank concentration of PAHs. The results were then applied to the blank correction. LOD and LOQ of the method were determined by PAHs concentration in 11 blank samples. The recovery test was conducted by analyzing unexposed commercial polyethylene (PE) pellets. A total of 1 g of PE pellets were spiked with a known amount of standard solution and left to dry in the desiccator. Other PE pellets were kept at their origin, treated as the real samples. The recoveries for PAHs ranged from 87.21 ± 5.15 to 100.48 ± 5.42.

## 3. Results and Discussion

### 3.1. Abundance and Distribution of Surface Water MPs from Canals to the Sea

MPs were detected at all sampling sites in this study with 3408 MPs pieces in total (canals—375 pieces, river—772 pieces, sea—2261 pieces), and average abundances of MPs in different sampling areas are presented in Table 1. In particular, the highest average abundance of MPs was in three canal systems (104.17 ± 162.44 pieces/m^3^), followed by Can Gio sea (2.08 ± 2.22 pieces/m^3^) and the lowest in the Saigon River (0.6 ± 0.38 pieces/m^3^) (Figure 2). The result shows the ubiquity of MPs in aquatic environments, which leads to a potential impact on living creatures and the ecosystem.

Among three canal systems, the abundance of MPs in NL-TN was the highest (165 ± 280.63 pieces/m^3^), followed by TH-BN (114.17 ± 46.45 pieces/m^3^), and TH-LG (33.33 ± 20.41 pieces/m^3^). In TH-BN, MPs abundance in most of sampling sites was higher 100 pieces/m^3^ (except TH-BN-3, 45.83 pieces/m^3^). TH-BN is a large canal of HCMC with 20 km in length stretching though central districts to Saigon River. The canal connects with other canals, such as TH-LG, and intersects with Saigon River. Therefore, pollutants including MPs from other canal systems could be emitted into TH-BN. The abundance of MPs was higher at two ends of the canal. At the beginning of TH-BN, there are still makeshift houses along the bank of the canal, meanwhile, at the downstream (central district of HCMC), there are many commercial activities such as tourist yachts, restaurants on the river, and water transportation. The reception of domestic wastewater from these places led to higher abundances in these sampling sites. NL-TN pours into Saigon River at NL-TN-5. Most of sampling sites had abundances lower than 60 pieces/m^3^, and the abundance of MPs tended to decrease from the upstream to the downstream of the canal (from 666.67 pieces/m^3^ to 25 pieces/m^3^). This can be explained: since NL-TN-1 is the starting point of the canal route, the water was stagnant and could not be circulated. Furthermore, domestic wastewater from the outlet of the sewage-collection system at the beginning of the canal might cause the increase of MPs abundance. TH-LG, located deep in the inner city, is the shortest canal among three canals, and there are mostly residential activities along the canal. This canal intersects with TH-BN at TH-LG-3, and the abundance of MPs increased at two end sampling sites of the canal (TH-LG-4 and TH-LG-5). Results of MPs in canals in this study were compared with similar sampling areas (Table 2): the MPs’ abundance were lower than that of Wuliangsuhai lake, China (3120–11,250 pieces/m^3^) [44] và Dongting Lake and Hong Lake, China (900–4650 pieces/m^3^) [45], while being higher than that of Lake Victoria, Uganda (0.02–2.19 pieces/m^3^) [46].

MPs from urban canals followed the current and were discharged into Saigon River which joins with Dong Nai River and flows into Can Gio Sea (Figure 1). Therefore, rivers are the main source for plastic discharge in the ocean [47]. The abundance of MPs in surface water of Saigon River was less than in three canal systems. The lowest abundance was at SG-7 (0.16 pieces/m^3^) and highest at SG-12 (1.60 pieces/m^3^). Lower abundance of MPs in Saigon River might be caused by the dilution of the higher flow of Saigon River in a larger area. In addition, some polymers have higher density (for instance: Polyethylene terephthalate, d = 1.38 g/mL) than fresh water (d = 1 g/mL); therefore, these polymers might deposit under the river bed. Furthermore, a portion of MPs may have penetrated into the food chain through aquatic animals [4,48,49,50,51]. The inconsistent MPs abundances among sampling sites in Saigon shows that, in addition to the water provided from canals, the surface water of Saigon River is also influenced by other anthropogenic activities. Saigon River is the waterway for ships entering or leaving Saigon’s ports, and cargo ships could incidentally discharge wastes, which could be another potential source of MPs. Along the river, higher abundances of MPs at some sampling sites (SG-1, SG-5, SG-8) compared to adjacent sites were observed near the locations receiving flow from inner city’s canals (Figure 1). In particular, SG-12 (1.59 pieces/m^3^)—the highest abundance of MPs site—not only receives the flow from TH-BN canal, but also experiences many riverside activities such as the riverside park and a harbor (one of the biggest harbors of HCMC). MPs abundance in river of this study was also compared with other research (Table 2). Compared to Cisadane River, Indonesia (13.33 to 113.33 particles/m^3^), Ganges River, India (38 ± 4 MP/m^3^), and Yellow River near its estuary, China (497,000–930,000 items/m^3^), the abundance of MPs in this study was lower. The abundance of MPs, on the other hand, was higher than that of Beijiang River, China (0.56 ± 0.45 items/m^3^). The above studies also stated that the possible origins of MPs may come from domestic wastewater, and anthropogenic activities on the river and along the river banks. Previous studies reported that MPs in road dust contributed to the accumulation of MPs in surface water through runoff (O’Brien et al., 2021; Vogelsang et al., 2018; Yukioka et al., 2020). In European countries, 50% of MPs in road dust derived from tires and road marking paint enter the environment every year, and MPs were also fragmented from plastic waste on the road such as packaging or plastic bottles under the impact of sun radiation or physical force (Monira et al., 2021). Therefore, the emission sources of MPs in canals and river in this study could be from residential activities, waterway transport, tourism along the canal, wastewater as well as road dust. The populations of urban districts are approximately 40,000 people/km^2^, showing the high potential for MP pollution.

Can Gio Sea receives not only inland water from Saigon River but also from other Rivers such as Long Tau, Vam Lang, and Dong Tranh. MPs abundance in Can Gio Sea varied from 0.42 pieces/m^3^ to 8.27 pieces/m^3^. Compared to Saigon River, MPs abundance in Can Gio Sea is higher, but lower than that in canals. It is interesting that even though coastal water bodies (e.g., bays and estuaries) receive the inland flow of rivers, these areas are more polluted than rivers [49]. This may be explained by the emission of multiple sources and circulation patterns there (semi-enclosed basins). MPs in the surface water of Can Gio sea were collected 500 m offshore; this water body not only receives MPs from the inland currents but also from other coastal activities—aquaculture, tourism, seaports, and seafood markets [50]. In sampling sites CG-1 to CG-3 located near the estuaries of Vam Lang River and Dong Tranh River, the abundance of MPs increased from site CG-1 to site CG-3 (from 1.18 to 4.81 pieces/m^3^) (Figure 2). The abundance of MPs at CG-3 was higher probably because, near this site, there is the popular seaside resort of Can Gio. In sites CG-4 to CG-9, MPs abundance was significantly lower than other sites (<1 pieces/m^3^). This could be explained by the farther sampling locations from the shore compared to other sites (Figure 1). MPs abundance at CG-8 was lowest, as this site was farthest from the coast. MPs abundance at CG-10 and CG-11 (especially CG-11, with 8.27 pieces/m^3^) was much higher than other adjacent sites, possibly because these two locations also intersect with the flow from Ghenh Rai Bay (Figure 1). Interestingly, the chemical composition results also show similarities in the distribution of MPs in Can Gio Sea. Particularly, nearshore sampling sites (CG-1 to CG-3) had fewer and consistent polymer types (Figure 3), while offshore sites (CG-4 to CG-9) had more diverse polymer types (Figure 3). The results show that MPs abundance in Can Gio Sea (2.08 ± 2.21 pieces/m^3^) was higher than that of the North-East Atlantic (0–1.5 items/m^3^) [51] and mid-North Pacific Ocean (0.51 ± 0.36 items/m^3^) [52]. However, MPs abundance near estuaries in other studies were reported to be higher than this study, for example: Sebou Estuary and Atlantic Coast, Morocco (10 to 168 particles/m^3^) [53] and Yangtze Estuary, China (0–259 items/m^3^) [54]. The higher abundance of MPs in the seawater near the estuaries than that of rivers was reported previously, for example: Xiangxi Bay contained 0.11–68 pieces/m^3^ compared to the Beijiang River of South China (0.56 ± 0.45 pieces/m^3^) [24]. Compared to other studies, MPs abundance in HCMC was relatively low. However, we focused on MPs from 0.5 mm to 5 mm; therefore, the abundance of smaller MPs (<0.5 mm) might be underestimated. In the report of Peter [55], research concerning MPs with sizes from 10 μm may have higher abundance, 1000 times than that of those investigating MPs from 100 μm. Different sampling methods would lead to inconsistent results. For example, using bucket or pump as samplers, the number of detected MPs was much higher than using a manta trawl; this result was stated in the research of Felishmino [56]. This can be explained by the larger mesh size of the net (manta trawl; 300 μm) leading to the loss of small MPs and fibrous MPs. Even the above comparisons might be inaccurate due to the differences in the studies’ MPs size, sampling and analysis methodologies, sampling time, etc. These research studies still contribute to the general overview of MPs pollution in surface water in the world [57]. Thus, in this study, microplastics were found in all surface water samples from canals, rivers, and oceans. The abundances of MPs varied within the same sampling area and in different sampling areas. The abundance of MPs tends to decrease gradually when transporting from canals to Saigon River and increasing in Can Gio Sea. This difference in abundance may be due to the influence of water flow and different emission sources on the water bodies and along the banks of canals, river, and the coast.

### 3.2. Physical Characteristics of Surface Water MPs

In this study, MPs from 0.5 mm to 5 mm were analyzed for their distribution in terms of size, shape, and color. MP size was classified into groups: from 0.5 mm to 1.0 mm, from 1.0 mm to 2.8 mm, and from 2.8 mm to 5 mm (Figure 4). The size distribution varied in different sampling areas. The three canal systems and Can Gio Sea showed the highest proportion of size class of 1–2.8 mm (40.8%—canal systems; 49.1%—Can Gio Sea) while 2.8–5 mm MPs were dominant in Saigon River (42.6%). On average, MPs smaller than 2.8 mm accounted for the majority in all sampling sites (69.4%). Granule, fragment, and fiber were three common shapes found in all sampling sites (Figure 5). However, their distribution was uneven among sampling areas (Figure 6). The results show that MPs in canals were mainly fiber (37.3%) while granule accounted for the largest proportion of MPs in Saigon River (43.5%). On the other hand, Can Gio Sea data show the highest percentage of fragments (48.4%). Eight different colors of MPs were detected, some common colors were white, transparent, green, and blue (Figure 7). There was a similarity in the distribution of color among sampling areas, as white was the dominant one, and the percentage was the highest in Can Gio Sea (52.1%).

In three canal systems, MPs larger than 1 mm accounted for the majority (40.8%), canals are the primary receiver of MPs from inland emission sources plastic wastes, and MPs have not yet experienced environmental effects (sun radiation, physical impacts of water, chemical, biological impacts) so they have not suffered much fragmentation. The high percentage of fibrous MPs may relate to the wash-holding process from domestic wastewater; this is considered as the main source of microfibers in the environment [62]. Studies worldwide also give similar results with fiber being the most dominant shape of MPs [62,63,64,65]. The study by Chenxi Wu stated that fiberous MPs tend to be more abundant in more populated areas [63].

In Saigon River, the proportion of granular MPs increased, as well as the size of MPs (2.8 mm to 5 mm). This result is reasonable because we found a high number of resin pellets in this sampling area. These pellets’ size were from 2.8 mm to 5 mm. HCMC is the industrial center of Vietnam with many plastic factories, resin pellets are regular imports to produce plastic items [64]. Saigon River is the waterway for ships entering and leaving Saigon ports, so cargo boats could incidentally discharge wastes, which could be another potential source of MPs. Other studies show that MPs found in rivers were fiber or fragment shape (Table 2). Concerned size of MPs were different in each study, however, the common MPs size tended to be inversely proportional to the number of MPs [47,48,63].

In Can Gio Sea, fragment was the dominant shape (48.4%). The sources of these MPs could be from agricultural activities along the coast where mulch films are utilized to conserve soil moisture [66]; aquacultural and tourism activities also contributed to the high proportion of fragmented MPs through the fragmentation of utensils and single-use plastic items. In addition, MPs in the sea experience more environmental impacts and exist for a longer period of time, making MPs more susceptible to fragmentation. Therefore, the highest proportion of fragment was rational. Compared to other published research, the most abundant MP shape found in seawater was also fragment (Table 2). This was quite reasonable because the majority of MPs in this study and other studies were Polyethylene (PE) and Polypropylene (PP). These polymers have lighter density (d = 0.9–1.0 g/mL) than seawater (d = 1.025 g/mL); therefore, they can easily float on the surface water. Mesh samplers (manta trawl, Hydro-bios trawl, …) could cause the loss of a number of fiberous MPs, which was stated in the research of Felismino [56]. This may also be the cause of the lower percentage of fiberous MPs in Saigon River and Can Gio Sea.

In this study, 6 different colors were detected in canals, whereas we found 8 different colors in Saigon River and Can Gio Sea (Figure 7). The number of colors in each sampling area also indicate the diversity of emission sources. White was the most common color and transparent was one of major colors in all sampling areas. Transparent MPs pieces appearing to be common in surface water possibly originated from the single-use packaging plastic products (containers, bottles, cups, or bags). This finding is suitable since white and transparent are the most common color manufactured [67].

Previous studies found that fish might mistakenly intake white, yellow, or brown MPs (as they closely resemble their zooplankton prey) [68], while sea turtles commonly ingest translucent and light-colored plastics [69]. Blue MPs were reported to have been accidently consumed by a variety of marine animals such as fishes [70,71] and clams [48]. Further research into the surface water in Ho Chi Minh City is needed to verify which among the attributes of MPs aid the ingestion of marine organisms.

### 3.3. Chemical Composition of Surface Water MPs

All MPs in canals were analyzed for chemical composition by FTIR-ATR. Due to the large number of MPs detected in Saigon River and Can Gio Sea, MPs were classified into groups by color and shape, and representatives of each group were selected to determine their polymer (328 spectrums for Saigon River, 382 spectrums for Can Gio Sea). In total, there were 13 polymers identified in all sampling areas, including 10 homopolymers—PP (Polypropylene); PE (Polyethylene); EVA (Ethylene-vinyl acetate); ABS (Acrylonitrile-butadiene-styrene); PS (Polystyrene); PET (Polyethylene terephthalate); Nylon (all polyamides); PC (Polycarbonate); PTFE (Polytetrafluoroethylene) or FEP (Fluorinated ethylene propylene); PMMA (Poly(methyl methacrylate))—and 3 copolymers—copolymer PP and PE; copolymer PP and EVA; copolymer PE and ABS (Appendix A).

In three canal systems, we detected 12 types of polymers, of which PP, PE, and EVA accounted for the highest proportion (64%). PC, PTFE, and PMMA were the minority of the total MPs (5%; 0.3%; and 2.1%, respectively). Figure 3a shows that 11 polymer types were found at NL-TN-1, where the stagnant water phenomenon occurs and is also a start of the canal. In Saigon River, 7 polymers were confirmed (PP, PE, EVA, ABS, PET, PP/PE, and PE/ABS), of which PP and PE were the two predominant polymers (74.1% in total) (Figure 3b). The percentage of PP was the highest (42.3%), PE was the second most popular polymer (31.9%), and they were detected at all sites. In addition, some polymers with relatively small percentages were found, such as ABS at SG-12, PET at SG-7, (PP/PE) and (PE/ABS) at SG-14, SG-15. There are similarities between the chemical composition of MPs in Saigon River and Can Gio Sea. The highest percentage of polymers belongs to EVA with 33.3%, then PP with 29.6%, and PE with 27.1% (Figure 3c). PS only appeared at 3/15 sites (0.2%). Other polymers (ABS, PC, PET, nylons, PP/PE, and PP/EVA) accounted for about 7.9%. PP and EVA were presented in all sampling sites, from sites CG-5 to CG-8, there were various polymers (5–7 types) while the other sites only observed 2 to 3 polymer types.

In this study, in terms of the abundance in collected samples, 3 types of plastic PE, EVA, and PP were high ubiquity due to the following reasons. On the one hand, these plastics have a buoyant density PE (0.88–0.96 g/cm^3^), EVA (0.937 g/cm^3^), PP (0.905 g/cm^3^) while the density of river water and seawater is from 1 to 1.025 g/cm^3^, which makes them tend to float on the water’s surface [1]. Moreover, as a result of Chubarenko’s research determining the properties and settling behaviors of MPs, we know that polyethylene fiber present in the euphotic zone from 6 to 8 months before they are sunk by the biofouling process [72]. In fact, more than half of the plastic produced globally is floating plastic, so the dominant occurrence of these polymers in the environment is understandable [73]. On the other hand, the distribution of MPs endures impacts due to wind redistribution, convergence zones, gyre entrainment, the activity of sediment microorganisms, and so on, which leads to MPs concentrating at some hot spots while another site may be insignificant [74].

Marine MPs accumulation, along with oceanographic processes (dense shelf water cascading, severe coastal storms, offshore convection, saline subduction, and the adhesion of plants and plankton) enhance the sedimentation ability of high-density plastics (ABS, PS, PET, PA, PC, PTFE or FEP, PMMA), which explains why their percentage in the collected samples was relatively low [75]. These MPs take less than 18 h to enter the marine sediments process [72]. PA’s origin comes from textiles, the automotive industry, carpets, kitchenware, and sportswear; PS is used in single-use products such as containers, lids, bottles, and trays [76]. PC is principally employed in electronics, data storage, and components in phones [77]. PET is used for products such as bags, packaging, wrappers, and engineering resins [78]. PMMA is used in a wide range of fields which include vehicles, lenses for glasses, and medical and dental applications [79]. PTFE, another name for Teflon, is most commonly used as a nonstick coating for cookware, but also for the manufacture of semiconductors and medical devices, and as an inert ingredient of pesticides [80].

Interestingly, copolymers (PP/PE, PP/EVA, and PE/ABS) were found in this study. The copolymer is produced from two or more different types of monomers resulting in a more complex structure to enhance the material properties. For instance, PP/PE is a material with low temperature resistance and improved tensile and toughness [1]. Fused filament fabrication (FFF), used in 3D printers, is made from ABS. However, it has poor thermal stability due to the thermo-oxidative degradation of butadiene monomers [81]. The combination of high-density PE (for instance, HDPE) and ABS increase mechanical strength (tensile, flexure, and compressive), which significantly improves material properties. Additionally, the recycling process of plastic waste also contributes to the creation of copolymers [82].

Overall, types of polymers have quite consistent distribution in all sampling areas due to the fact that the distribution of buoyant plastics had more dominance than high-density plastic in surface water. The results of this study were compared with previous reports in the same region and around the world, which resulted in a similar tendency. PP, Nylon 6, and PE accounting for 49.06% were found in the Chao Phraya River, Thailand [83]; the percentage of PP, PE, and PS was 93.66% in the Pearl River, China [84]; in three rivers in southeastern Norway (Akerselva river, Hobøl river, and Gryta river), two types of plastic (PP and PE) were from 52.9% to 85.7%, respectively [85]. MPs in surface water and sediment was dominated by PE fragments (53–67%) followed by PP (16–30%) and PS (16–17%) in the Bay of Brest, Brittany, France [86]. These results reflect the huge production and consumption of PP and PE globally [87].

### 3.4. Characteristics of PAHs Associated with of Surface Water MPs from Canals to the Sea

In this section, we report the concentrations of 14 PAHs in MPs. Appendix A shows the levels of MPs-bound PAHs collected in the canals (*n* = 15), Saigon River (*n* = 15), and Can Gio Sea (*n* = 15). Overall, PAHs were detected in most of all MPs samples in 3 sampling areas. Total PAHs in MPs found in Can Gio Sea had the highest concentration (1398.99 ± 1612.14 ng/g), followed by that collected in canals and finally in Saigon River with a concentration of 1070.34 ± 1613.93 ng/g, and 926.68 ± 695.55 ng/g, respectively. This result suggests that there were more PAHs accumulating in MPs in Can Gio Sea. The canals carrying MPs flow into Saigon River and finally enter Can Gio Sea. During the transportation, MPs experienced exposure to PAHs, leading to the absorption and the enrichment of these compounds.

In the canals, the total concentration of 14 PAHs absorbed by MPs ranged from 30.94 to 8940.99 ng/g, with the average concentration being 1070.34 ± 1613.93 ng/g. Phe (3-ring PAH), BbF and InP (larger molecular weight PAHs compounds—5, 6 rings) had high concentration in MPs. BbF had the highest concentration of 225.6 ± 272.14 ng/g in MPs in the canals. The total concentration of PAHs in MPs in Saigon River varied from 432.95 to 3267.88 ng/g. Phe, AnT, and Pyr were dominant in MPs collected in Saigon River with values of 109.7 ± 142.98, 186.28 ± 101.35, and 174.94 ± 100 ng/g, respectively. Appendix A shows the concentration of PAHs associated with MPs, which was collected in several types of environments from previous studies. Our results were consistent with the study of Tan [24] in freshwater. In the Feilaixia reservoir, the concentration of MPs-bound PAHs with 3 and 4 rings, such as Phe and Chr, were higher than other PAHs. PAHs with 5 and 6 rings, for instance BghiP and InP, tend to have higher concentrations than the rest. Furthermore, DahA (5-ring PAH) was found at low or undetectable concentration [24]. The concentration of PAHs bound to MPs in Can Gio Sea (232.71–6448.66 ng/g) was lower than the results in the coastal area of Huanghai Sea and Bohai Sea, China [37], Zhengmingsi Beach and Dongshan Beach, China [88]. On the other hand, the PAHs concentration in this study was higher than that of Seal beach, USA, Thinh Long and Tonking Bay beaches, Vietnam [89], and the Southwest coast of Taiwan [10].

Figure 8 illustrates the proportion of PAHs by molecular weight including lower molecular weight PAHs (LMW-PAHs) (Ace, Flu, Phe, and Ant), medium molecular weight PAHs (MMW-PAHs) (Flt, Pyr, BaA, and Chr), and high molecular weight PAHs (HMW-PAHs) (BbF, BkF, BaP, DahA, BghiP, and InP). There was a difference of PAHs on MPs in the sampling areas in this study. HMW-PAHs were dominant in MPs found in canals, accounting for 63% of the total PAHs, whereas, in Saigon River, MMW-PAHs were found to have the highest concentration with a percentage of 57.5%. The concentration of LMW, MMW, and HMW- PAHs in MPs were similar in Can Gio Sea. MPs with granule shape accounted for a large number of MPs collected in Saigon River. In these granules, a pellet is an original form of commercial polymer. The adsorption capacity of pellets was smaller than other forms (foam, fiber) of MPs because of their small contact areas. The study of Van [20] found that high concentrations of PAHs were found in commercial PS foam (240–1700 ng/g) and a lower one in commercial PS pellets (12–15 ng/g). This is the reason why PAHs concentration found in MPs in rivers were smaller than in canals and sea in this study. We also analyzed concentrations of PAHs in the original PE pellet and found concentrations of total PAHs ranging from 25.85 to 104.85 ng/g. Moreover, 3 and 4 ring PAHs, for instance Flu, Phe, Flt, and Pyr were dominant in PE pellets. This could explain the dominance of MMW-PAHs in Saigon MPs samples.

We applied the ratio of PAH isomers to diagnose the sources of PAHs. The ratios of Flu/(Flu + Pyr) and Ant/(Ant + Phe) have been commonly used to examine the sources of PAHs [10,23,39]. The Ant/(Ant + Phe) ratio is <0.1, and Flu/(Flu + Pyr) ratio is <0.4 indicating that PAHs were mainly contributed by petrogenic sources. Vice versa, if these values are larger than 0.1 and 0.4, respectively, PAHs were primarily from pyrogenic sources [23,40]. Figure 9 shows the diagnostic ratios of PAHs on MPs samples collected in the three sampling areas. There were 9 of 15 MPs samples found in Can Gio Sea that had a Flu/(Flu + Pyr) ratio < 0.1 and Ant/(Ant + Phe) ratio < 0.4. This result reveals that PAHs on MPs in Can Gio Sea were mainly from petrogenic sources. Our results are consistent with many previous studies about PAHs on MPs on the sea surface water or coast in previous studies (Appendix A). It is very likely that MPs on the sea water surface are exposed to floating oil leaking from ships. PAHs associated with MPs on 6 samples found in Saigon River, were also strongly affected by PAHs from petrogenic sources. Moreover, PAHs on MPs in canals and river were also from a mix of petrogenic and pyrogenic sources. The HMW-PAHs are good indicators for vehicular emission sources [88]. Particularly, BghiP and InP are considered to represent a maker for diesel vehicles [10]. Low-ring PAHs are primarily derived from petroleum sources [88]. The HMW-PAHs are good indicators for vehicular emission sources [90]. In this study, the high level of HMW-PAHs associated with MPs in canals reflects vehicle emission source of PAHs in MPs. In addition, the canals in HCMC are the main system of rainwater discharge that carries particles, road dust from the air and on the streets into the marine environment. Many studies about PAHs in road dust showed that HMW-PAHs were the most abundant PAHs [91,92]. On the other hand, MPs were also reported to have high concentration in road dust and could be possible carriers for PAHs [93]. Nevertheless, the study of Ida [94] also reported high MPs concentration on roads and in nearby stormwater, sweep sand, and wash water. Tire and road wear particles have been identified as a potential major source of MPs in road dust and stormwater and could contribute to the abundance of MPs in the aquatic environment [66,94].

## 4. Conclusions

This study contributed to our understanding of the abundance and physical and chemical compositions of MPs, as well as provided the first information of PAHs associated with MPs in the surface water (canals, river, and sea) in Ho Chi Minh City, Vietnam. MPs were found at all sampling sites, with the highest abundance found in the urban canals (104.17 ± 162.44 pieces/m^3^). The abundance of MPs decreased from canals to Saigon River and increased in Can Gio Sea. The three common shapes were fragment, fiber, and granule, and each shape was predominant in a specific sampling area. White accounted for the highest proportion of all MPs’ colors (34.7%–52.1%). In this study, 13 types of polymers (including 3 co-polymers) were confirmed by FTIR-ATR, of which PE, PP, and EVA were three dominant polymers. Total PAHs in MPs found in Can Gio Sea had the highest concentration (1398.99 ± 1612.14 ng/g), followed by that collected in canals and finally in Saigon River with concentration of 1070.34 ± 1613.93 ng/g, and 926.68 ± 695.55 ng/g. PAHs on MPs in Can Gio Sea were mainly from petrogenic sources due to oil leaking from ships. Vehicle emission sources could be a significant pyrogenic source of PAHs in MPs in canals and rivers. This study provides basic data on the fate of MPs in the surface water of HCMC for future investigations. This is also the first study on PAHs associated with MPs in Vietnam.

## Figures and Tables

**Figure 1 polymers-14-02450-f001:**
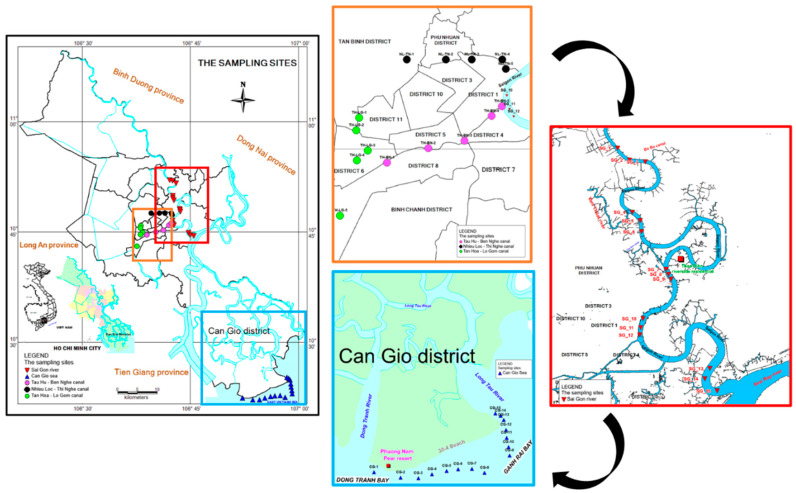
Sampling map of surface water MPs in HCMC, Vietnam, in August 2020.

**Figure 2 polymers-14-02450-f002:**
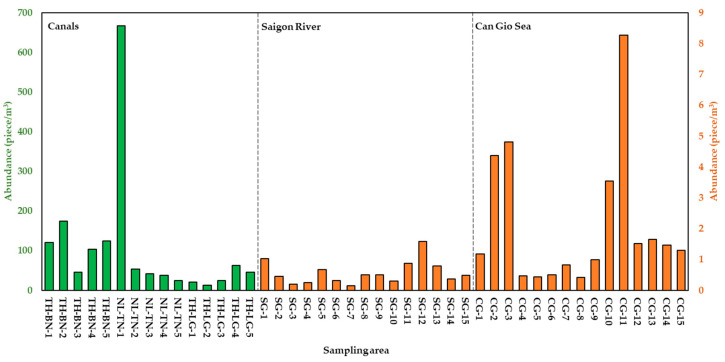
Abundance of MPs in canal systems, Saigon River, and Can Gio Sea in HCMC, Vietnam, in August 2020.

**Figure 3 polymers-14-02450-f003:**
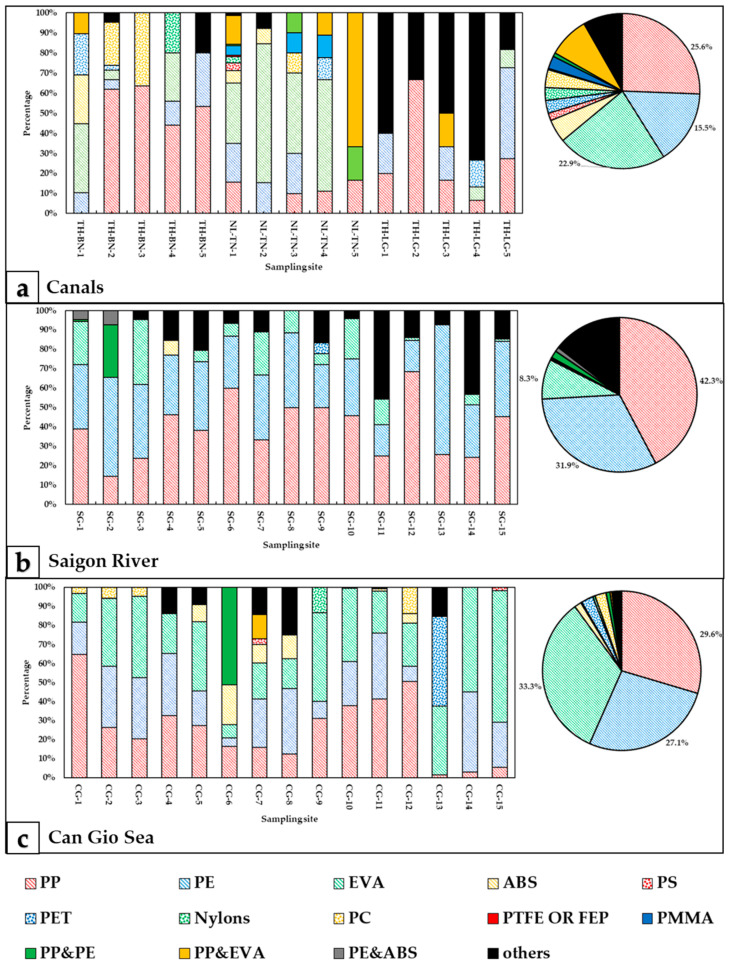
Chemical composition of MPs: (**a**) Canals, (**b**) Saigon River, and (**c**) Can Gio Sea, in August 2020.

**Figure 4 polymers-14-02450-f004:**
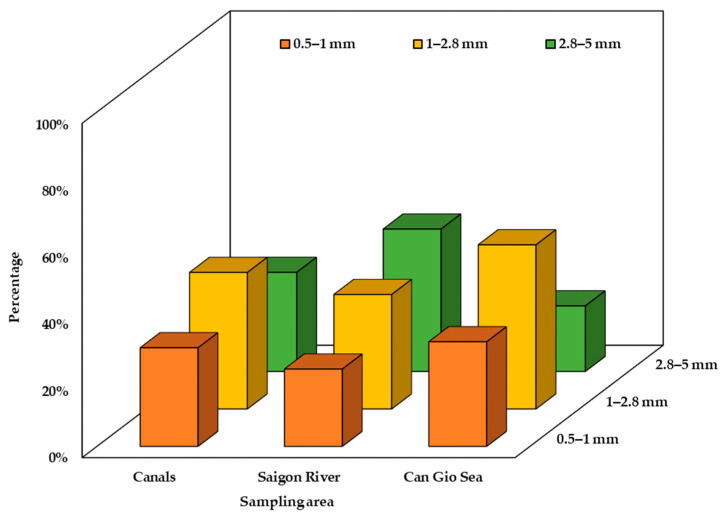
Size distribution of MPs in canal systems, Saigon River, and Can Gio Sea, in August 2020.

**Figure 5 polymers-14-02450-f005:**
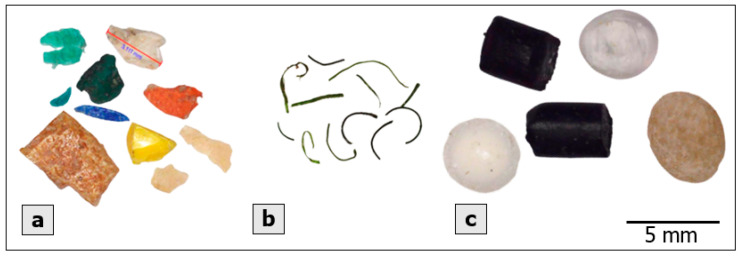
Typical microplastics of different shapes of this study: fragment (**a**), fiber (**b**), granule (**c**).

**Figure 6 polymers-14-02450-f006:**
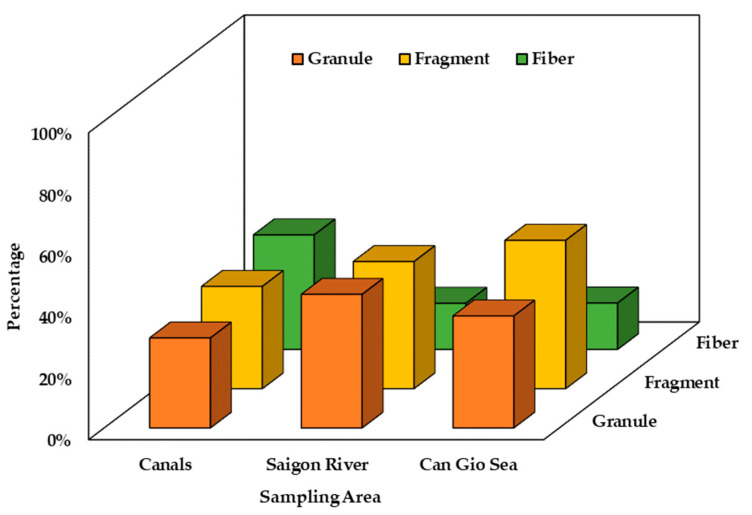
Shape distribution of MPs in canal systems, Saigon River, and Can Gio Sea, in August 2020.

**Figure 7 polymers-14-02450-f007:**
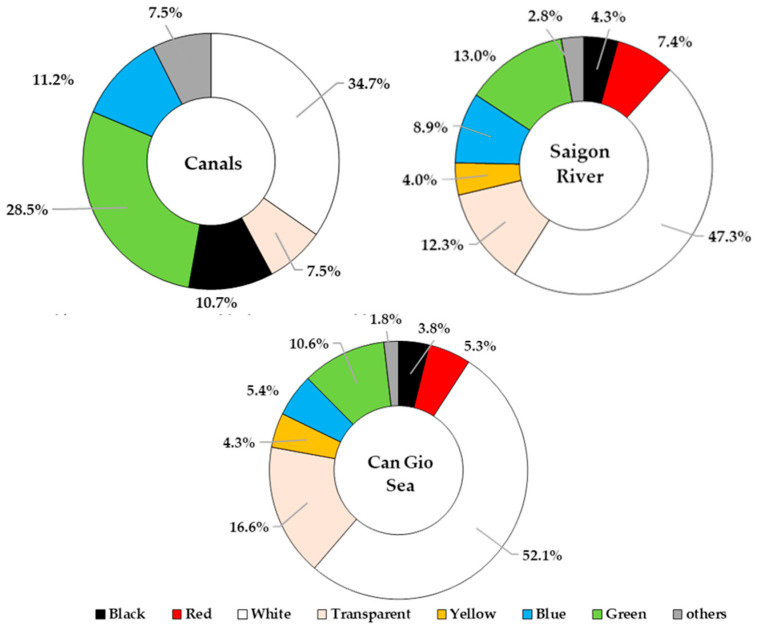
Color distribution of MPs in canal systems, Saigon River, and Can Gio Sea, in August 2020.

**Figure 8 polymers-14-02450-f008:**
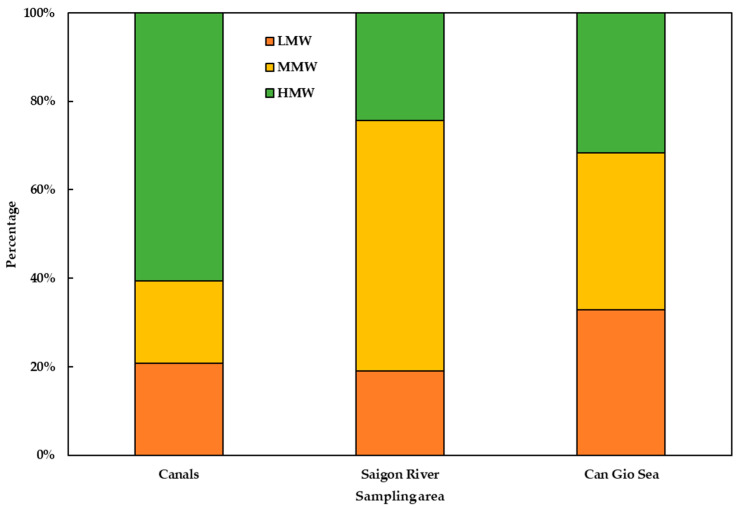
Percentage of PAHs bound to MPs by molecular weight in canals, Saigon River, and Can Gio Sea in August 2020.

**Figure 9 polymers-14-02450-f009:**
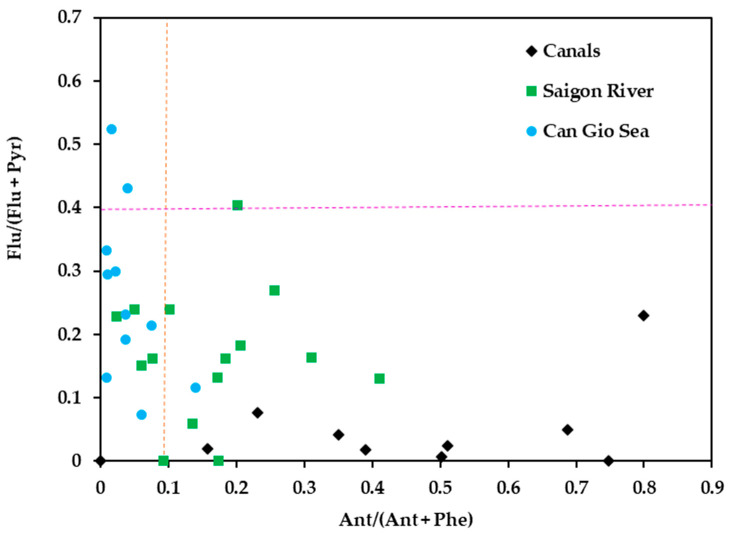
The bi-diagnostic ratios of Flu/(Flu + Pyr) and Ant/(Ant + Phe) in MPs found in Canals (black diamond), Saigon River (green square), and Can Gio sea (blue dot).

**Table 1 polymers-14-02450-t001:** Mean abundance of MPs in canal systems, Saigon River, and Can Gio Sea.

Sampling Areas	MPs Abundance (Pieces/m^3^)		
	Max	Min	Mean
Canals	666.67	12.50	104.17
Saigon River	1.59	0.16	0.60
Can Gio Sea	8.27	0.42	2.08

**Table 2 polymers-14-02450-t002:** Microplastics in surface water of previous studies.

No.	Region	Sampler	Abundance	Common MPs Size	Common MPs Shape	Common MPs Color	Common MPs Composition
1	Wuliangsuhai Lake, northern China [44]	Stainless steel bucket	3120–11,250 pieces/m^3^	<2 mm (98.2%)	Fiber	No data	PS, PE
2	Dongting Lake and Hong Lake, China [45]	12 V DC Teflon pump	900–4650 pieces/m^3^	<2 mm (65%)	Fiber	Transparent, blue	PP, PE
3	Lake Simcoe, Ontario, Canada [56]	Low volume grabs and manta trawls	0–700 particles/L (grab), 0.4–1.3 particles/m^3^ (manta trawl)	No data	Fiber (grab) Fragment (manta trawl)	No data	PP, PE
4	Lake Victoria, Uganda [46]	Manta trawl	0.02–2.19 pieces/m^3^	<1 mm (36%)	Fragment (36.7%)	White/transparent (59.1%)	PE, PP
5	Three canal systems of HCMC, Vietnam (this study)	Stainless steel bucket	104.17 ± 162.44 pieces/m^3^	1.0–2.8 mm	Fiber	White, transparent, blue, green	PP, PE, EVA
6	Cisadane River, Indonesia [58]	Stainless-steel bucket	13.33 and 113.33 particles/m^3^	0.5–1.0 mm	Fragment	No data	PE, PS, PP
7	Beijiang River, China [24]	Plankton net (mesh size, 0.112 mm and diameter, 20 cm)	0.56 ± 0.45 items/m^3^	0.6–2 mm	Film	No data	PP, PE
8	Ganges River, India [59]	Hand- operated bilge pump	38 ± 4 MP/m^3^	2.459 ± 0.209 mm average	Fiber	blue	Rayon
9	Yellow River near estuary, China [60]	Stainless steel bucket	497,000–930,000 items/m^3^	<0.2 mm	Fiber	No data	PE, PS, PP
10	Snake and Lower Columbia rivers, USA [61]	Grab and net	0 to 13.7 MPs/m^3^	<0.5 mm	Fiber	No data	PE, PP, PET
11	Saigon River, HCMC, Vietnam (this study)	Hydro-bios trawl	0.60 ± 0.38 pieces/m^3^	2.8–5.0 mm	Granule/Pellet	White, transparent, blue, green	PP, PE
12	Yangtze Estuary, China [54]	Metal cylinder	0–259 items/m^3^	<1mm (79%)	Fragment	White and transparent	PE, PP, α-cellulose
13	Sebou Estuary and Atlantic Coast, Morocco [53]	Steel sampler	10 to 168 particles/m^3^	0.1–0.5 mm	Fragment	While and blue	No data
14	North-East Atlantic [51]	Manta trawl	0–1.5 items/m^3^	1.00–2.79 mm	Fragment (63%)	White, transparent, and black	No data
15	Mid-North Pacific Ocean [52]	Manta trawl	0.51 ± 0.36 items/m^3^	1.0–2.5 mm	Fragment (31%)	White and transparent	PP (53%)
16	Can Gio Sea, HCMC, Vietnam (this study)	Hydro-bios trawl	2.08 ± 2.22	1.0–2.8 mm	Fragment	White, transparent, blue, green	PP, PE, EVA

## Data Availability

Not applicable.

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
