# Peer review of "Characteristics of Microplastics and Their Affiliated PAHs in Surface Water in Ho Chi Minh City, Vietnam"

_polymers, 2022, doi:10.3390/polym14122450_

Round 1
Reviewer 1 Report
This manuscript investigated the distribution of microplastics and PAHs associated with MPs in three main canal systems, Saigon River, and Can Gio Sea of Ho Chi Minh City (HCMC). The objectives of this study are: 1) to provide the information on the abundance and distribution of MPs in surface water in canals, rivers and sea in HCMC; 2) to characterize the physical and chemical composition of MPs; 3) to investigate the distribution of 14 PAHs affiliated to MPs and their potential sources. This facilitates a better understanding of MPs occurring in urban rivers in the developing country and their potential adsorption effect of PAHs. I would recommend acceptance after major revision according to the following comments.
1. Abstract: line 11, affiliated means adsorption? The authors mentioned “adsorption" in the introduction part (line 51). Affiliation includes physical attachment, chemical adsorption, etc., such as van der Waals force. In the method part, the author described that PAH is not included in the dissolution liquid from H2O2 treatment (line 163). So, clearly that it remains a discrepancy. The authors should clarify the definition and use words clearly.
2. It seems that the urban canals, Sai Gon River, and Can Gio sea have a relationship of upstream and downstream. Thus, I am curious why the authors did not provide information about the impact of such a relationship on spatial discrepancies of microplastic occurrence and the adsorption of PAHs. To see if the stories in these three areas are potentially related. It can be an interesting point.
3. The authors should add PAH concentration data obtained from these surface waters.
4. According to the map, line 13, Can Gio Sea is this an estuarine zone? It should be more accurately described.
5. Line 46, water environment is the sink of plastic garbage
6. Line 49, the authors should elaborate on the toxicity risk of PAH to show the most basic implications of this study. “PAHs in MPs may be from manufacture or adsorbed from the environment.” you may need literature evidence;
7. Lines 62-72, the authors should describe the potential scale of PAHs’ occurrence in Vietnamese water bodies, e.g., corroborated by source data or industrial land uses, to suggest the significance of studying microplastic & PAHs adsorption in this region, but not just because that this topic has not been studied here before.
8. Line 89, the entire study appears to have been sampled only once, in August of 2020, and thus how to draw conclusions of general significance
9. Figure 1. the fonts on the map are too small
10. Lin 117, the individual differences of microplastics are often significant, so when the samples are divided into two groups here, how to ensure that the microplastic characteristics and adsorbed PAH of the two groups are the same and can be compared equally, and the random differences of their groupings do not affect the study results
11. Line 160, what are the criteria for determining a specific type of plastic material
12. In Section 3.1, the text uses the term "probably" in many places when analyzing why the abundance of MPs in the water column is too high or too low, but does not explain clearly why this is the reason for the change in abundance.
13. Figure 2, clearly identifies the Y-axis of each panel
14. Page 11, line 354, “Concerned size of MPs……the number of MPs.” The article does not analyze the relationship between microplastic size and quantity, nor does it mention the relevant literature, so how to reach this conclusion
15. Figure 8, molecular weight, PAH? MPs? Why not show PAH data in the text, which is the core of this article
16. What is the statistical link between the characteristics and abundance of MPs and PAHs
17. Page 18, lines 513-517, ..applied the ratio of PAH isomers to diagnose the sources of PAHs. What are the criteria and basis for distinguishing sources?
18. Page 18, lines 524-531. The data analysis was incomplete. The follow-up analysis only mentioned the source of HMW-PAHs, and the sources of other types of PAHs were not discussed. But the article elaborated during the analysis that “in Saigon River, MMW-PAHs were found to have the highest concentration with the percentage of 57.5%.”
19. Page 18, lines 532-534, “Tire and road wear …… the aquatic environment.” Reference?
Author Response
The Editor of Polymers,
Dear Ms. Summer Zhou,
Title: Characteristics of microplastics and their affiliated PAHs in surface water in Ho Chi Minh City, Vietnam
Ref.: polymers-1756634
Thank you for your letter dated 27 May 2022, together with the comments on our manuscript. We would like to acknowledge you and the reviewers for the valuable and informative comments. We have carefully considered the comments and have addressed them in the following attachment.
In response to the comments, we have added information to the manuscript. All authors have agreed with the changes.
We think that the comments are very helpful to improve the quality of the paper. We hope that our response is satisfactory to you and the reviewers, and the manuscript is acceptable for resubmission and publication in the Polymers. Next, we would provide detailed responses to comments.
Sincerely,
To Thi Hien and co-authors
University of Science, Vietnam National University Ho Chi Minh City
Comments and Suggestions for Authors
We thank both reviewers for their careful comments. Below we include the reviewers’ comments (in black) and our responses to them (in blue).
Reviewer #1:
This manuscript investigated the distribution of microplastics and PAHs associated with MPs in three main canal systems, Saigon River, and Can Gio Sea of Ho Chi Minh City (HCMC). The objectives of this study are: 1) to provide the information on the abundance and distribution of MPs in surface water in canals, rivers and sea in HCMC; 2) to characterize the physical and chemical composition of MPs; 3) to investigate the distribution of 14 PAHs affiliated to MPs and their potential sources. This facilitates a better understanding of MPs occurring in urban rivers in developing country and their potential adsorption effect of PAHs. I would recommend acceptance after major revision according to the following comments.
- Abstract: line 11, affiliated means adsorption? The authors mentioned “adsorption" in the introduction part (line 51). Affiliation includes physical attachment, chemical adsorption, etc., such as van der Waals force. In the method part, the author described that PAH is not included in the dissolution liquid from H2O2 treatment (line 163). So, clearly that it remains a discrepancy. The authors should clarify the definition and use words clearly.
Response:
Thank you for your comment.
- “Affiliated” does mean adsorption. This word is widely used for the adsorption of persistent organic compounds (POPs) on MPs (Mai et al. 2018; Chen et al. 2020). One of the reasons why MPs affiliate with POPs (in this case: PAHs) is the relatively large surface area of MPs (physical attachment). Moreover, the hydrophobic component of MPs interacts with hydrophobic organic compounds such as PAHs (chemical adsorption) (Hirai et al. 2011; Chen et al. 2018). So, we can confirm that the word “affiliated” was used correctly.
- We want to clarify that the H2O2 treatment is used for removing natural organic matters (not POPs) on MPs. The study of Mai et al., 2018 (Mai et al. 2018) has provided evidence that pretreatment in 20 mL of 30% H2O2 did not significantly affect PAHs concentrations on MPs. That means PAHs are not decomposed or dissolved in H2O2 and they are still affiliated with MPs.
Reference:
Chen CF, Ju YR, Lim YC, et al (2020) Microplastics and their affiliated PAHs in the sea surface connected to the southwest coast of Taiwan. Chemosphere 254:126818. https://doi.org/10.1016/J.CHEMOSPHERE.2020.126818
Chen R, Qi M, Zhang G, Yi C (2018) Comparative experiments on polymer degradation technique of produced water of polymer flooding oilfield. In: IOP Conference Series: Earth and Environmental Science. p 12208
Hirai H, Takada H, Ogata Y, et al (2011) Organic micropollutants in marine plastics debris from the open ocean and remote and urban beaches. Mar Pollut Bull 62:1683–1692. https://doi.org/10.1016/j.marpolbul.2011.06.004
Mai L, Bao LJ, Shi L, et al (2018) Polycyclic aromatic hydrocarbons affiliated with microplastics in surface waters of Bohai and Huanghai Seas, China. Environ Pollut 241:834–840. https://doi.org/10.1016/J.ENVPOL.2018.06.012
- It seems that the urban canals, Sai Gon River, and Can Gio sea have a relationship of upstream and downstream. Thus, I am curious why the authors did not provide information about the impact of such a relationship on spatial discrepancies of microplastic occurrence and the adsorption of PAHs. To see if the stories in these three areas are potentially related. It can be an interesting point.
Response:
Thanks for your comment. We explained the difference in distribution of MPs and adhered PAHs that have a relationship of upstream and downstream in section 3.4 (lines 506 to 517, page 17). In summary, our result suggests that there were more PAHs accumulating on MPs in Can Gio Sea. The canals carrying MPs flow into Sai Gon River and finally enter Can Gio Sea. During the transportation, MPs experienced the exposure to PAHs, the absorption, and the enrichment of these compounds. However, more research on PAHs in microplastics in those water bodies is needed to get more convincing evidence of this relationship.
- The authors should add PAH concentration data obtained from these surface waters.
Response:
Thanks for your comment. We have already presented PAHs concentrations on MPs in surface waters in Table S3. PAHs concentration was associated with MPs (mean ± SD, min-max, unit: ng/g) in three aquatic environments in HCMC.
- According to the map, line 13, Can Gio Sea is this an estuarine zone? It should be more accurately described.
Response:
Thank you for your suggestion. We rewrote the study area in the manuscript “the sampling area included the estuarine zone of rivers from inland” (line 119, page 3).
- Line 46, water environment is the sink of plastic garbage
Response:
Thank you for your clarification.
- Line 49, the authors should elaborate on the toxicity risk of PAH to show the most basic implications of this study. “PAHs in MPs may be from manufacture or adsorbed from the environment.” you may need literature evidence;
Response:
The authors thank the reviewer, we modified in the manuscripts: “PAHs are well-known because of their toxicity, carcinogenicity, and mutagenicity [22], [23]. Besides, MPs can adsorb and concentrate PAHs from the aquatic environments, then transfer and bioaccumulate through the food chain and finally affect human health”. (Line 51 to 54; page 2)
We added reference [20] and [21] for the statement “PAHs in MPs may be from manufacture or adsorbed from the environment” (Line 49 to 50; page 2)
- Lines 62-72, the authors should describe the potential scale of PAHs’ occurrence in Vietnamese water bodies, e.g., corroborated by source data or industrial land uses, to suggest the significance of studying microplastic & PAHs adsorption in this region, but not just because that this topic has not been studied here before.
Response:
Thank you so much for the reviewer's suggestion. The studies of PAHs in water bodies in Vietnam are mainly focusing on sediment instead of the water phase. Therefore, the authors have added the level of PAHs in sediment in Sai Gon River and PAHs in the manuscript.
“In Vietnam, previous studies have found a remarkable level of PAHs in water bodies specifically in sediment in HCMC [37], [38]. The concentration of total 16 PAHs in sediment in Sai Gon River ranged from 49 ng/g to 933 ng/g dw, with the dominance of Phe, Flt, Pyr, and BbF [38]. It is also important to understand the level of MPs affiliated PAHs because MPs are one of the chemical carryings and can transfer throughout the food web [24], [39], [40]” (lines 71 to 76, page 2).
- Line 89, the entire study appears to have been sampled only once, in August of 2020, and thus how to draw conclusions of general significance.
Response:
In this study, we investigated 3 sampling areas with 15 samples/area (45 samples in total). For each sampling area, we chose equally spaced sampling sites to ensure that the samples are representative of the study area. Some previous studies also conducted sampling once (Wang et al. 2020; Haddout et al. 2021). In addition, in this study, due to limited funding, we only conducted sampling once in August 2020 (rainy season). Therefore, further studies should be conducted to clarify seasonal variation of MPs and affiliated PAHs.
Reference:
Haddout S, Gimiliani GT, Priya KL, et al (2021) Microplastics in Surface Waters and Sediments in the Sebou Estuary and Atlantic Coast, Morocco. Anal Lett 55:256–268. https://doi.org/10.1080/00032719.2021.1924767
Wang S, Zhang C, Pan Z, et al (2020) Microplastics in wild freshwater fish of different feeding habits from Beijiang and Pearl River Delta regions, south China. Chemosphere 258:127345. https://doi.org/10.1016/J.CHEMOSPHERE.2020.127345
- Figure 1. the fonts on the map are too small
Response:
Thanks for your comment. We changed the fonts on the map (Figure 1)
- Line 117, the individual differences of microplastics are often significant, so when the samples are divided into two groups here, how to ensure that the microplastic characteristics and adsorbed PAH of the two groups are the same and can be compared equally, and the random differences of their groupings do not affect the study results.
Response:
In this study, we want to investigate MPs and PAHs affiliated with MPs at a same sampling site, therefore, we divided the amount of sample into two parts to ensure the existence of PAHs affiliated with MPs in the collected sample. To minimize the certain errors due to uneven distribution of MPs in surface water, we shook the sample to mix it well before dividing into two parts.
- Line 160, what are the criteria for determining a specific type of plastic material
Response:
In this study, the identification for specific polymers is based on a standard spectrum database shown in Melissa’s research (Jung et al. 2018). The IR spectrum of collected plastics was compared to the IR polymers’ standard spectrum at each peak site which is representative for function groups (absorption bands (cm− 1)) of specific polymers. Reference:
Jung MR, Horgen FD, Orski S V., et al (2018) Validation of ATR FT-IR to identify polymers of plastic marine debris, including those ingested by marine organisms. Mar Pollut Bull 127:704–716. https://doi.org/10.1016/j.marpolbul.2017.12.061
- In Section 3.1, the text uses the term "probably" in many places when analyzing why the abundance of MPs in the water column is too high or too low, but does not explain clearly why this is the reason for the change in abundance.
Response:
In this study, we focused on MPs in surface water, we did not investigate MPs in water columns therefore we do not have more explanation in the manuscript. The term “probably” was used once (line 287, page 7) in section 3.1.
- Figure 2, clearly identifies the Y-axis of each panel
Response:
Thank you for your clarification. We changed the Y-axis of each panel in the manuscript (Figure 2, line 213, page 6)
- Page 11, line 354, “Concerned size of MPs……the number of MPs.” The article does not analyze the relationship between microplastic size and quantity, nor does it mention the relevant literature, so how to reach this conclusion.
Response:
Thank you for your comment, we added references for the statement “From the previous studies’ results, concerned size of MPs was different in each study, however, the common MPs size tended to be inversely proportional to the number of MPs [47], [48], [63], [68]” (lines 361 to 362, page 11).
- Figure 8, molecular weight, PAH? MPs? Why not show PAH data in the text, which is the core of this article
Response:
Thank you for your suggestion. The PAH data was shown in Table S3. And, we added some more PAH data in the text:
“In the canals, the total concentration of 14 PAHs absorbed on MPs ranged from 30.94 to 8,940.99 ng/g, with the average concentration was 1,070.34 ± 1,613.93 ng/g. Phe (3-ring PAH), BbF and InP (larger molecular weight PAHs compounds - 5, 6 rings) had high concentrations on MPs. BbF had the highest concentration of 225.6 ± 272.14 ng/g on MPs in the canals. The total concentration of PAHs in MPs in Sai Gon River varied from 432.95 -to 3,267.88 ng/g. Phe, Flt, and Pyr were dominant in MPs collected in Sai Gon River with the value of 109.7 ± 142.98, 186.28 ± 101.35, and 174.94 ± 100 ng/g, respectively” (lines 573 to 579, page 16).
- What is the statistical link between the characteristics and abundance of MPs and PAHs
Response:
Thanks for your comment. In this study, we have not found the clear correlation between MPs’ characteristics and PAHs. Therefore, there is a need to have further studies to give clearer evidence of the relationship between MPs characteristics and PAHs abundance.
- Page 18, lines 513-517, ..applied the ratio of PAH isomers to diagnose the sources of PAHs. What are the criteria and basis for distinguishing sources?
Response:
Thank you for reviewer’s question. The criteria for distinguishing sources are shown on Figure 9 (the dashed line). We also added this information in the text: “We applied the ratio of PAH isomers to diagnose the sources of PAHs. The ratios of Flu/(Flu + Pyr) and Ant/(Ant + Phe) have been commonly used to examine the sources of PAHs [10], [23], [39]. The Ant/(Ant + Phe) ratio is < 0.1, and Flu/(Flu + Pyr) ratio is < 0.4 indicating that PAHs were mainly contributed by petrogenic sources. Vice versa, if these values are larger than 0.1 and 0.4, respectively, PAHs were primarily from pyrogenic sources [23], [40]. Figure 9 shows the diagnostic ratios of PAHs on MPs samples collected in the three sampling areas” (lines 615 to 621, page 18)
- Page 18, lines 524-531. The data analysis was incomplete. The follow-up analysis only mentioned the source of HMW-PAHs, and the sources of other types of PAHs were not discussed. But the article elaborated during the analysis that “in Saigon River, MMW-PAHs were found to have the highest concentration with the percentage of 57.5%.”
Response:
Thank you for the reviewer’s comment. The author provided more source information of other types of PAHs in the text:
“The HMW-PAHs are good indicators for vehicular emission sources [91]. Particularly, BghiP and InP are considered to represent as a maker for diesel vehicles [40]. Low-ring PAHs are primarily derived from petroleum sources [91]” (lines 534 to 537, page 18)
In Sai Gon River, the diagnostic ratios showed that both petrogenic and pyrogenic sources are principal sources of PAHs on MPs. The high concentration of MMW-PAHs on MPs in Sai Gon River because MPs with granule shape accounted for a large number of MPs collected in Sai Gon River. We also analyzed concentrations of PAHs on the original PE pellet and found concentrations of total PAHs ranging from 25.85 to 104.85 ng/g. Moreover, 3 and 4 ring PAHs, for instance, Flu, Phe, Flt, and Pyr were dominant on PE pellets. This reason could explain the dominance of MMW-PAHs in Saigon MPs samples.
- Page 18, lines 532-534, “Tire and road wear …… the aquatic environment.” Reference?
Response:
Thanks for your comment. The authors add references for this statement (Reference [66], [97]; line 642, page 18)
“Nevertheless, the study of Järlskog [96] also reported high MPs concentration on roads and in nearby stormwater, sweep sand and wash water. Tire and road wear particles have been identified as a potential major source of MPs in road dust and stormwater and could contribute to the abundance of MPs in the aquatic environment [66], [97]”.

Reviewer 2 Report
The authors conducted a valuable monitoring program. Reasonable method and huge workload make this study have reference value. There are some concerns where minor revisions are needed:
1. Figure 2, The authors claim that they have parallel samples, but with no error bars in Fig.
2. I recognize the significance of this study, but the authors' statements in this section can be improved.
3. Line 234, I prefer a table to compare the different abundance between rivers. Just like Table 2.
4. Maybe authors should add a “the” before “canals”.
5. Figure 4, are they collected from the study area? Some of them do not looks like plastics, FTIR spectrum should be afflicted to them.
6. The discussion should focus on the authors' statements, and the references are used to prove them. So, don't review the published studies and use them.
Author Response
The Editor of Polymers,
Dear Ms. Summer Zhou,
Title: Characteristics of microplastics and their affiliated PAHs in surface water in Ho Chi Minh City, Vietnam
Ref.: polymers-1756634
Thank you for your letter dated 27 May 2022, together with the comments on our manuscript. We would like to acknowledge you and the reviewers for the valuable and informative comments. We have carefully considered the comments and have addressed them in the following attachment.
In response to the comments, we have added information to the manuscript. All authors have agreed with the changes.
We think that the comments are very helpful to improve the quality of the paper. We hope that our response is satisfactory to you and the reviewers, and the manuscript is acceptable for resubmission and publication in the Polymers. Next, we would provide detailed responses to comments.
Sincerely,
To Thi Hien and co-authors
University of Science, Vietnam National University Ho Chi Minh City
Comments and Suggestions for Authors
We thank both reviewers for their careful comments. Below we include the reviewers’ comments (in black) and our responses to them (in blue).
Reviewer #2:
The authors conducted a valuable monitoring program. Reasonable methods and huge workload make this study have reference value. There are some concerns where minor revisions are needed:
- Figure 2, The authors claim that they have parallel samples, but with no error bars in Fig.
Response:
Thanks for your comment. In this study, we collected 45 samples (15 samples in each sampling area - canals, Saigon River, and Can Gio River). Figure 2 presents the abundance of MPs in each sample so there is no error bar in Figure 2. In addition, we also presented the average abundance of each sampling area in the text, lines 206-209, page 5.
- I recognize the significance of this study, but the authors' statements in this section can be improved.
Response:
Thank you for your comment.
- Line 234, I prefer a table to compare the different abundance between rivers. Just like Table 2.
Response:
We mentioned the abundance of MPs of other rivers in the manuscript, lines 260-265, page 7. In addition, Table 2 includes the information of MPs in different sampling areas (canals, rivers, and oceans) in the World. We clarified the sampling area of published studies in Table 2.
- Maybe authors should add a “the” before “canals”.
Response:
Thank you for your suggestion.
- Figure 4, are they collected from the study area? Some of them do not looks like plastics, FTIR spectrum should be afflicted to them.
Response:
All microplastics in figure 4 were collected from our study areas. We recorded their appearance with a digital microscope and their images were archived for the purpose of describing their shape, size and color. We confirmed the identity of these MPs pieces by FTIR-ATR.
- The discussion should focus on the authors' statements, and the references are used to prove them. So, don't review the published studies and use them.
Response:
Thanks for your comment.

Round 2
Reviewer 1 Report
As authors have responded in detail to the review comments, I suggest accepting the manuscript